# Twin boundary migration in an individual platinum nanocrystal during catalytic CO oxidation

Jérôme Carnis [1,2,8], Aseem Rajan Kshirsagar [3], Longfei Wu [1,2], Maxime Dupraz[1,2], Stéphane Labat[1], Michaël Texier[1], Luc Favre [1], Lu Gao[4], Freddy E. Oropeza[4], Nimrod Gazit[5], Ehud Almog[5], Andrea Campos[6], Jean-Sébastien Micha[7], Emiel J. M. Hensen [4], Steven J. Leake[2], Tobias U. Schülli[2], Eugen Rabkin [5], Olivier Thomas [1], Roberta Poloni[3], Jan P. Hofmann [4,9] & Marie-Ingrid Richard [1,2,10 ✉]

At the nanoscale, elastic strain and crystal defects largely influence the properties and functionalities of materials. The ability to predict the structural evolution of catalytic nano-crystals during the reaction is of primary importance for catalyst design. However, to date, imaging and characterising the structure of defects inside a nanocrystal in three-dimensions and in situ during reaction has remained a challenge. We report here an unusual twin boundary migration process in a single platinum nanoparticle during CO oxidation using Bragg coherent diffraction imaging as the characterisation tool. Density functional theory calculations show that twin migration can be correlated with the relative change in the interfacial energies of the free surfaces exposed to CO. The x-ray technique also reveals particle reshaping during the reaction. In situ and non-invasive structural characterisation of defects during reaction opens new avenues for understanding defect behaviour in confined crystals and paves the way for strain and defect engineering.

[1] Aix Marseille Université, Université de Toulon, CNRS, IM2NP, Marseille, France. [2] ID01/ESRF, The European Synchrotron, Grenoble, France. [3] Grenoble-INP, SIMaP, University of Grenoble-Alpes, CNRS, Grenoble, France. [4] Laboratory for Inorganic Materials and Catalysis, Department of Chemical Engineering and Chemistry, Eindhoven University of Technology, Eindhoven, The Netherlands. [5] Department of Materials Science and Engineering, Technion-Israel Institute of Technology, Haifa, Israel. [6] Aix Marseille Univ, CNRS, Centrale Marseille, FSCM (FR1739), CP2M, Marseille, France. [7] CRG-IF BM32 beamline at the European Synchrotron (ESRF), CS40220, Grenoble Cedex 9, France. [8] Present address: Deutsches Elektronen-Synchrotron (DESY), Hamburg, Germany. [9] Present address: Surface Science Laboratory, Department of Materials and Earth Sciences, Technical University of Darmstadt, Darmstadt, Germany. [10] Present address: Univ. Grenoble Alpes, CEA Grenoble, IRIG, MEM, NRS, Grenoble, France. ✉email: mrichard@esrf.fr

Platinum (Pt) nanoparticles (NPs) are widely used catalysts in many important fields like the chemical industry, fuel cell technology, and automobile exhaust gas purification[1–3]. Of particular interest for catalysis is the simultaneous in situ characterization of the chemical, morphological, and structural (strain, facets, and defects) dynamical[4] evolution of individual nanoparticles in (near) operational conditions. Recently, it has been observed that strain, as well as defects are key factors for catalysis, opening-up the concept of mechanochemistry, and strain-engineered catalysis[5–8]. Applying a strain to metal catalysts, either compression or tension, can change the way they perform[6]. Recently, microstrain or localized lattice strain, which is generated from structural defects, such as dislocations, grain boundaries, and twin boundaries, has been observed to enhance catalytic activity and stability[9–11]. In particular, multitwin defects, which serve as catalytically active sites, have been proven to promote electrocatalytic activity[12–16]. Twinned structures may form during crystal nucleation, growth, phase transformations, recrystallization[17], or via the mechanism of deformation twinning, concurrently with the plastic deformation by slip and for metals exhibiting a low stacking-fault energy[18]. Crystal twinning is also sensitive to the chemical environment. The adsorption of impurity atoms may contribute to the formation of twins[19]. Whereas twinning mechanisms are well known, descriptions of detwinning mechanisms have been relatively sparse. They occur mostly in shape memory alloys[20–22] or under a mechanical load[23,24]. To improve our understanding of physical defect mechanisms and enable defect engineering, it is crucial to develop imaging techniques capable of resolving defects and their evolution. Recently, it has been demonstrated that Bragg coherent diffraction imaging (BCDI) is particularly sensitive to defects and can resolve their three-dimensional (3D) structure nondestructively[25–27]. By illuminating an isolated object by an X-ray beam with coherence lengths larger than the object, and measuring its oversampled 3D diffraction pattern, it is possible to reconstruct the measured complex-valued object using phase retrieval iterative algorithms[28,29]. Its amplitude is related to the real space density or more specifically to Bragg electron density of the contributing crystal, whereas the phase is equivalent to the displacement field component projected on the Bragg diffraction vector. These two parameters can be resolved with a spatial resolution better than 10 nm[30–32].

In this article, we demonstrate the capabilities of BCDI to reveal in 3D the local morphology (faceting), strain and defect evolution in a single Pt nanoparticle in situ during catalytic CO oxidation reaction at elevated temperatures (450 °C and 500 °C) under Ar, CO, and/or $O_2$ atmosphere at near atmospheric pressure flow conditions.

## Results

Figure 1a shows the experimental setup. The sample consists of Pt nanoparticles obtained by solid-state dewetting on a sapphire substrate (see Methods), the Pt [111] direction being normal to the substrate surface. A lithographic processing route ensured that a number of Pt particles were well-separated from their neighbors that only one crystallite is irradiated by the incoming X-ray beam, and that the irradiated particle can be easily located after the experiment (see inset of Fig. 1a). The sample was placed in a reactor[33] with tunable parameters: temperature and gas conditions (see Fig. 1a and Methods). The BCDI measurements were performed by recording the intensity distribution in the vicinity of the specular **111** Pt Bragg reflection, which yields, for example, the 3D diffraction pattern displayed in Fig. 1b, the duration of the measurement being around 4 min 30 s The phase retrieval procedure used for spatial reconstruction of the

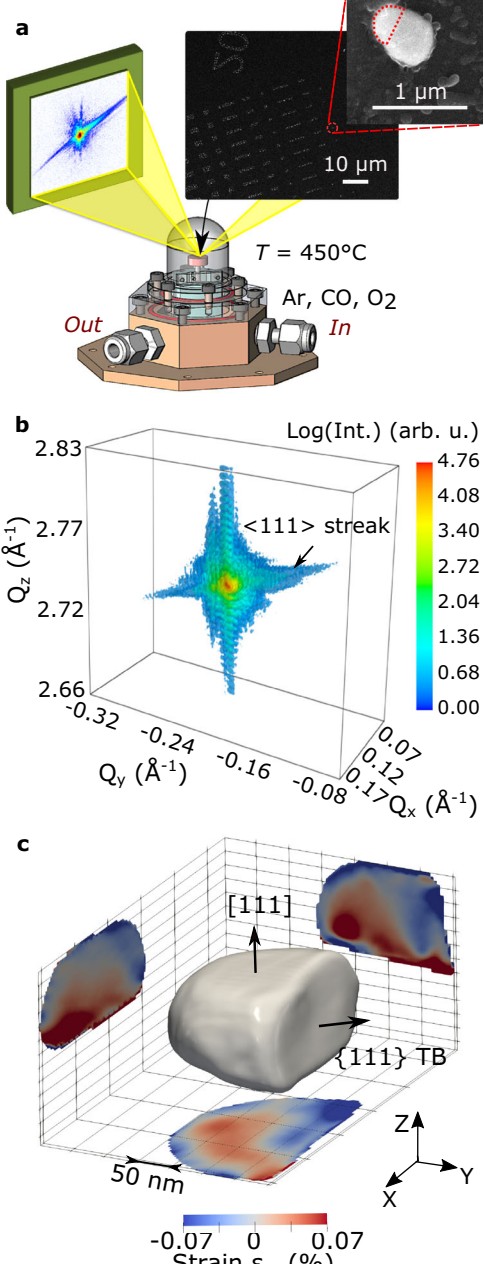

**Fig. 1 Experimental setup and measured individual particle in reciprocal and real spaces. a** View of the patterned sample placed in the reactor. The red dotted semi-circle on the scanning electron microscope (SEM) image indicates the measured crystal grain. **b** Measured 3D diffraction pattern at the 111 Pt Bragg peak in CO (2% in Ar) at 450 °C as a function of the reciprocal space coordinates ($Q_x$, $Q_y$, and $Q_z$). **c** Reconstructed modulus and strain (at the center of the crystal), with the presence of a {111} twin boundary (TB).

measured particle, as well as experimental details are described in Methods. The reconstructed Pt crystal (morphology and phase distribution) under CO (2% in Ar) exposure and at 450 °C is shown in Fig. 1c. The particle is approximately 200 x 320 x 110 (height) $nm^3$ in size and shows a complex structure, including a twin fault characterized by a {111} twin boundary (TB), in agreement with the <111> streak observed in the 3D diffraction pattern displayed in Fig. 1b. Σ3{111} twin boundaries (TBs) are commonly observed in face-centered cubic (f.c.c.) structures[34,35]. It corresponds to a boundary with a {111} composition plane

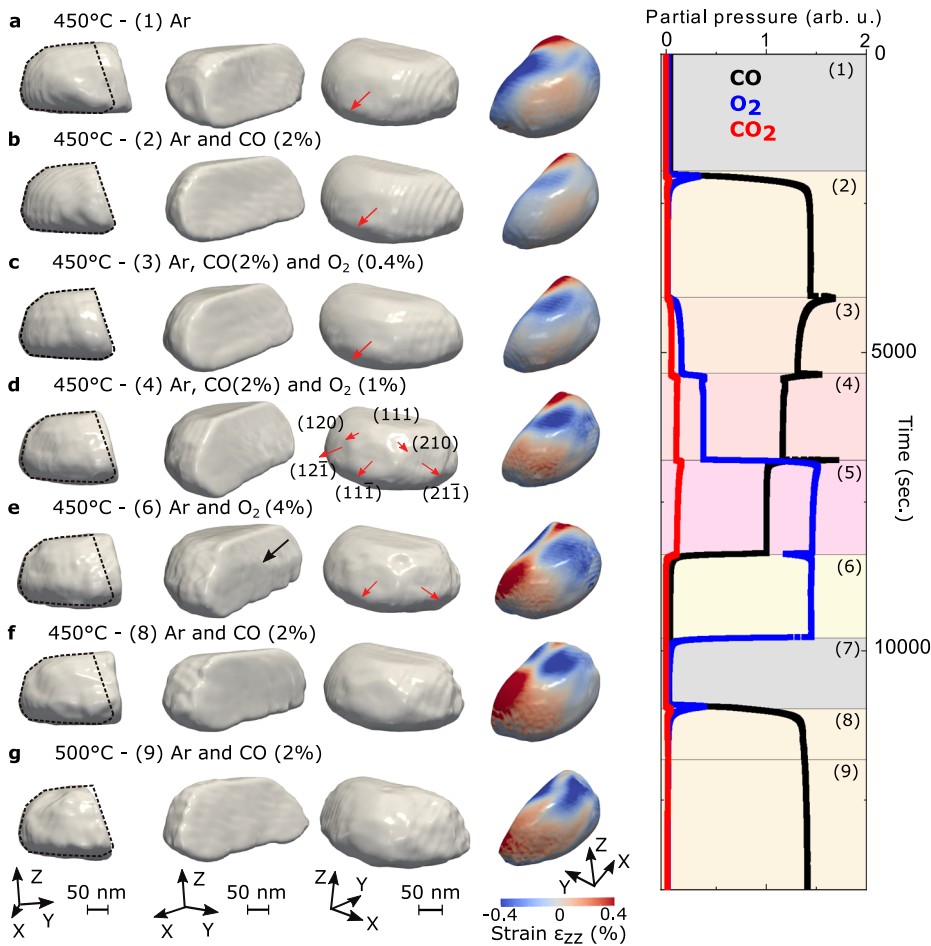

**Fig. 2 Evolution of the twinned Pt particle.** Left: 3D isosurface of the reconstructed modulus/electron density and strain field along the [111] vertical direction, $\varepsilon_{zz}$, drawn at 23% of the maximum density for each measured gas composition (at 450 °C under **a** Ar, **b** Ar and CO (2%), **c** Ar, CO (2%) and $O_2$ (0.4%), **d** Ar, CO (2%) and O2 (1%), **e** Ar and O2 (4%), **f** Ar and CO (2%) and at 500 °C under **g** Ar and CO (2%)). The black dotted contour is a guide for the eyes and delimits the shape of the crystal measured at condition (2): Ar and CO (2%) at 450 °C. The black arrow in Fig. **e** indicates the change of orientation of the interface. The red arrows show some of the facets of the crystal. Some of the facets are indexed. Right: Mass spectrometer signal during the experiment: CO (black), $O_2$ (blue), and $CO_2$ (red). The numbers between parentheses indicate the gas conditions (see Table S1).

(here, $(\bar{1}\bar{1}1)$ as fixed arbitrarily) separating two domains: the twin and its parent. The crystallographic orientation of the twin domain is obtained by a rotation of 180° around the $[\bar{1}\bar{1}11]$ direction of the parent grain. For instance, the (111) planes of the parent grain correspond to (115) planes in the twin grain[36] (see Figure S1). As the twin crystal planes are rotated with respect to the parent crystal lattice, they appear as a missing Bragg electron density at the **111** Pt Bragg reflection and thus as a void in the reconstructed modulus of the particle[37]. This gives the peculiar shape of the reconstructed particle, which is displayed in Fig. 1c, where only the parent grain with its $(\bar{1}\bar{1}1)$ TB is observed. Using a scanning X-ray diffraction microscopy approach[38] and owing to the mask processing route employed in sample preparation, it was then possible to identify the target Pt nanoparticle, as well as the measured domain (here, named as the parent grain), as demonstrated in the scanning electron microscopy (SEM) image of the very same particle displayed in Fig. 1a.

We collected sets of 3D coherent **111** Pt Bragg diffraction patterns at 450 and 500 °C for different gas compositions. The gas condition at near atmospheric pressure was changed in the following sequence at 450 °C: (1) pure Ar, (2) CO (2% in Ar), (3) CO (2% in Ar) + $O_2$ (0.4% in Ar), (4) CO (2% in Ar) + $O_2$ (1% in Ar), (5) CO (2% in Ar) + $O_2$ (4% in Ar) (data not saved), (6) $O_2$ (4% in Ar), (7) Ar (data not saved), and (8) CO (2% in Ar)

(see Table S1 and Fig. 2). The temperature was then increased to 500 °C under (9) CO (2% in Ar). BCDI measurements were performed twice after each change of the gas composition. Diffraction patterns of the **111** Pt Bragg peak for different gas compositions at 450 °C are shown in Figure S2. The intensity distribution shows interference arising from single-particle coherent diffraction. Clear changes in the reciprocal space intensity distribution (see the evolution of the inclination of the tilted streak) indicate modifications of the strain and/or shape of the measured catalyst under different gas atmosphere conditions. The real space 3D structure of the Pt crystal was reconstructed by the same iterative phase retrieval approach (see Methods) as previously. Figure 2 shows the evolution of the reconstructed modulus (or Bragg electron density) and of the strain along the [111] direction (normal to the substrate surface), $\varepsilon_{zz}$, of the catalyst in real space drawn at 23% of the maximum density[39] for each measured gas composition. The estimated experimental resolution ranges from 11.6 to 18.9 nm depending on the gas composition and temperature (see Figure S3 and Methods). The reconstructed real space data shows a clear evolution of the shape of the Pt crystal during CO oxidation reaction and temperature change. The black dotted contour is a guide for the eye and delimits the shape of the parent crystal measured at condition (2): Ar and CO (2%) at 450 °C (see also Fig. 1c). The estimated

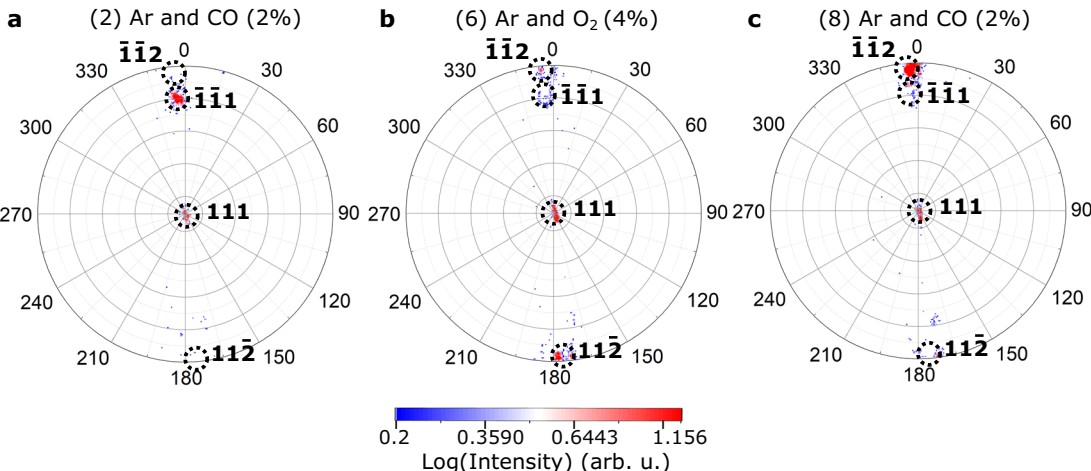

**Fig. 3 Faceting evolution.** Stereographic projections (pole figures) extracted from the 3D diffraction patterns at a Q value of 0.75 nm⁻¹ from the central **111** Bragg peak and integrated over $\Delta Q = 0.005$ nm⁻¹ and measured for different gas compositions at 450 °C: **a** for Ar + CO (2%), *i.e.*, condition (2) **b** for O$_2$ (4% in Ar), *i.e.*, condition (6) and **c** for CO (2% in Ar), *i.e.*, condition (8). Only the intense streaks of the diffraction patterns are observed. The dashed circles indicate the location of the peaks. They are guide for the eye.

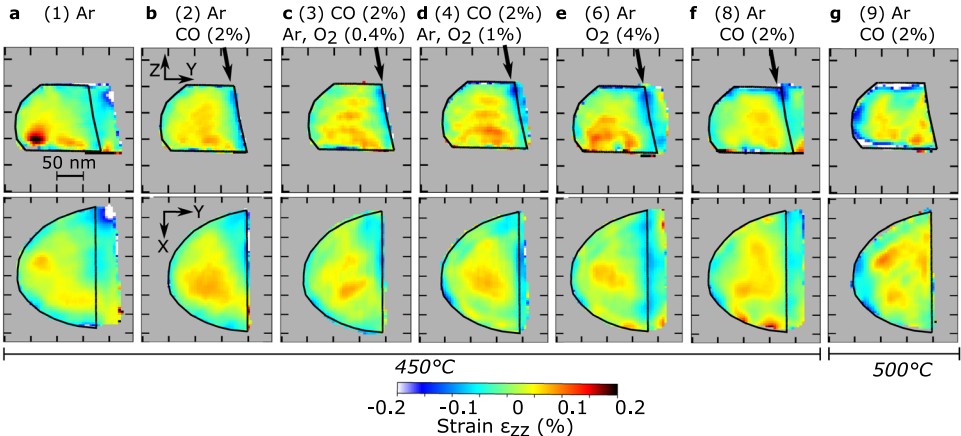

**Fig. 4 Out-of-plane strain evolution.** Out-of-plane strain maps at the center of the reconstructed crystal (at 450 °C under **a** Ar, **b** Ar and CO (2%), **c** Ar, CO (2%) and O2 (0.4%), **d** Ar, CO (2%) and O2 (1%), **e** Ar and O2 (4%), **f** Ar and CO (2%) and at 500°C under **g** Ar and CO (2%)). The dotted black contours are guide for the eyes and delimit the shape of the crystal measured at condition (2): Ar and CO (2%) at 450 °C. Tick spacing corresponds to 50 nm.

reconstructed crystal's volume is displayed as a function of time and gas mixture in Figure S4. At the beginning of the experiment (see Fig. 2a), the volume of the crystal is larger and shows a {111}-type twin boundary, perpendicular to the [1̄1̄1] direction. After introducing CO, a significant decrease in the crystal's volume is observed. The crystal still exhibits a [1̄1̄1] TB. This crystallite size decrease is caused by twin boundary migration. The twin crystal planes are rotated with respect to the parent crystal lattice and appear as a missing Bragg electron density, leading to a truncated shape of the measured crystal. By changing the gas mixture and introducing O$_2$ in an increasing ratio (from condition (3) to condition (6)), an inverse phenomenon, i.e., the increase of the crystallite's volume was observed. The increase of the crystal's volume through TB migration is the signature of the detwinning of the crystal. The parent crystal grows at the expense of the twin crystal. This was still observed when removing O$_2$ and introducing CO (see condition (8) in Fig. 2f). Interestingly, the TB does not migrate back to the initial configuration (i.e., condition (1)); it reaches a vertical grain boundary (GB). To determine the orientation of this grain boundary, we plotted the stereographic projection[40] of the 3D diffraction pattern corresponding to condition (8), as well as other most representative conditions

(conditions (2) and (6)) in Fig. 3. Stereographic projection is a convenient way to determine the orientation of facets and planar defects giving rise to streaks in 3D diffraction patterns. For all projections, the central spot corresponds to the top (111) surface. A 1̄11 bright spot, which corresponds to the [1̄1̄1] TB is observed for condition (2). It evolves toward a 1̄1̄2 spot (see Fig. 3c), i.e., a (1̄1̄2) interface/boundary. The GB observed in conditions (6) and (8) is then perpendicular to the [1̄1̄2] direction.

For the last condition (i.e., condition (9)), the temperature had been increased to 500 °C. This resulted in a decrease of the crystal volume through twin boundary migration (twinning) (see Fig. 2g). The crystal returned to the configuration observed at 450 °C under the same gas mixture (CO (2% in Ar), see condition (2)) and exhibits a [1̄1̄1] TB. Figure 2 also demonstrates that faceting occurs during reaction. Small facets of {111}, {210}, and {211}-types are observed for conditions (4) and (6). Note that these facets are too small (signal not intense enough) to be observed on the stereographic projections of Fig. 3.

The reconstruction results of the Bragg electron density (or modulus), the phase and the out-of-plane strain ($\varepsilon_{zz}$) of the Pt crystal are shown in Figures S5, S6, and 4, respectively, for the different gas mixtures. Two-dimensional slices through the center

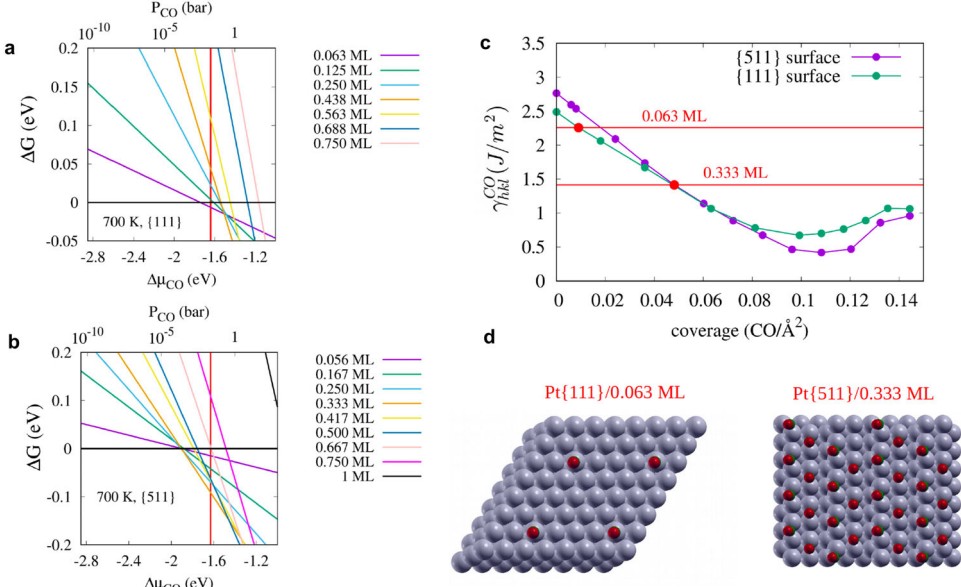

**Fig. 5 Interfacial energy.** Computed Gibbs free energy of adsorption per Pt atom on the surface, $\theta_{CO}\Delta G$ (eV/Pt atom), with $\theta_{CO}$ being the number of adsorbed molecule with respect to the full coverage case, at 700 K as a function of the chemical potential for **a** the {111} surface and **b** the {511} surface. The two vertical red lines correspond to $\Delta\mu_{CO}$ at 0.02 bar and 700 K with respect to 0 K. **c** Evolution of the interfacial energy $\gamma_{hkl}^{CO}$ as a function of CO coverage (*i.e.*, number of CO molecules adsorbed per Å²) for (111) and (511) surfaces. **d** The coverage predicted for the two surfaces are shown as horizontal red lines and the corresponding configurations are reported in d).

of the reconstructed crystal are displayed and the reconstructed crystal has been rotated in-plane for easier visualization, so that the twin interface is parallel to one axis. The evolution of the Bragg electron density and phase (proportional to the displacement field along the [111] direction, $u_z$) is well observed during reaction. As an example, the phase increases at the top of the crystal during detwinning: from condition (2) to (8), see Fig. S6. Local variations of the strain are also observed. After twinning (see Fig. 4b), the compressive strain was observed at the new twin boundary. During detwinning (from Figs. 4c–f), compressive strain remains at the position of the TB observed in Fig. 4b.

## Discussion

Interestingly, twinning is highly unfavorable in Pt, since the stacking-faults and twin boundary (322 mJ/m²) energies are high[41], making the nucleation of twins difficult. Twinning is seldom observed in Pt bulk crystals; however, it has been reported in nanocrystals[42–44], where a new twinning pathway has been identified[44]. It has been further observed that the formation of a twin boundary is a probable final configuration of sintered nanoparticles, even if the material shows high stacking-fault energies[45,46]. The latter case may explain the occurrence of a twin boundary in the Pt nanocrystal studied in this work.

Twinning/detwinning in f.c.c. crystals result from the emission of Shockley partial dislocations on successive atomic layers in {111} planes. These partial dislocations can be nucleated from grain boundaries (GBs)[47], free surfaces, from defects, such as ledges on TBs[48] or be multiplied by dislocation-twin boundary interaction[49]. Twinning/detwinning is generally dependent on: (a) sufficiently high local stress to trigger nucleation of twinning partials on successive planes, and (b) energetically favored transformation[50]. In our study, twinning was observed, while exposing the Pt crystal to CO gas flow, i.e., for conditions (2) and (9) (see Fig. 2 and S5). While the parent crystal is [111]-oriented normal to the substrate, the twin crystal is [115]-oriented[36]. When twinning occurs, the ratio of (115) to (111) exposed surface-area increases.

Adsorption of CO on Pt surfaces has received continued interest ever since Irving Langmuir studied this system[51] and the strong dependence of the CO adsorption energy on the surface structure is well established[52,53]. We have performed DFT calculations to compute the Pt/CO interface energy, $\gamma_{hkl}^{CO}$, (see Eqs. 1–5 of Methods) for both the {111} and {511} surfaces as a function of CO coverage. The configurations of CO molecules adsorbed on the {111} surface are taken from Ref. [53], where an extensive computational screening of many possible configurations for different coverages is reported. The reported lowest-energy configuration[53] at a given coverage is adopted here to compute the $\gamma_{hkl}^{CO}$. Due to the lack of previous computational studies of CO adsorption on Pt{511}, we computed several possible configurations at different coverages and considered the lowest-energy case to compute the Pt{511}/CO interface energy (see Figures S6-S21). The $\gamma_{111}^{CO}$ and $\gamma_{511}^{CO}$ computed by using the revPBE functional[54] are reported in Fig. 5. This functional was developed to improve the description of molecules adsorbed onto metallic surfaces and gives adsorption energies at low coverages of 137–140 kJ/mol (see Fig. S23), in good agreement with experimentally reported values[55–57] ranging between 126 and 145 kJ/mol. While for zero and low coverage, the Pt{111}/CO interface energy is lower than the {511} case, at high coverages (>0.07 molecule/Å² or 0.47 ML) the {511} is preferred. At intermediate coverages, between 0.04 and 0.07 molecule/Å², the difference in interfacial energy is negligible (see Fig. 5). Several exchange and correlations functionals were tested and found to yield slightly different numbers, as expected from previous works[51,58], but the same qualitative behavior for the two surfaces is predicted (see Table S3 and Figure S8). The CO adsorption is always stronger on {511} than {111}, regardless of the coverage, and for both surfaces it decreases with increasing CO coverage (see Figures S8 and S10). In order to establish the CO coverage expected for {511} and {111} at the experimental conditions of pressure and temperature, we computed the Gibbs free energy of adsorption as a function of the CO chemical potential for the lowest-energy configurations, see Fig. 5a–b. At 700 K and at a pression of 0.02 bar of CO, a coverage of 0.062 ML is predicted

for the {111} facets, while 0.333 ML would adsorb onto the {511} facets, yielding a difference in $\gamma_{hkl}^{CO}$ of 0.84 J/m². At 300 K and 0.02 bar, we predict a coverage of 0.56 ML for {111} (see Fig. S23), in good agreement with recent computational studies by Sautet et al.[59] and Gunasooriya et al.[53] The larger coverage predicted for the stepped {511} surface, as compared to the flatter {111} surface, is consistent with results from electron energy loss spectroscopy experiments[60].

The typical driving force associated with grain boundary migration in Pt can be estimated considering the driving force for recrystallization. It has been shown that the recrystallization in heavily cold-rolled Pt foils begins at temperatures as low as 400 °C[61]. The driving force of recrystallization, $E_{rec}$, is mainly related to the energy of stored dislocations, and can be roughly estimated as: $E_{rec} = 0.5\rho G b^2$. The maximum dislocation density in heavily deformed metals with high stacking-fault energy is of the order of $\rho \approx 10^{15}$ m⁻². For Pt, the magnitude of the Burgers vector is $b = 0.277$ nm, and the shear modulus, $G$, can be estimated by its isotropic value of 62 GPa. Then, $E_{rec} \approx 2.4 \times 10^6$ J/m³. On the other hand, the driving force for twin boundary migration in our working geometry is $E_s = \Delta\gamma/h$, where $\Delta\gamma = 840$ mJ/m² is the difference of the Pt/CO interface energies, and $h \approx 110$ nm is the particle height. The last equation can be derived by considering the infinitesimal lateral displacement of the boundary. With these data, $E_s \approx 7.6 \times 10^6$ J/m³. The two driving forces are of the same order of magnitude. Therefore, the energy gain from CO adsorption on the {511} facets represents a viable driving force for twin boundary migration. Another example of a small difference of surface energies driving the grain boundary migration is abnormal grain growth in thin films. Indeed, Humphreys and Hatherly[62] have shown that only 2% difference in surface energies of two neighboring grains can drive the abnormal growth. Note that, before the BCDI measurements presented in this manuscript, the sample was already at 450 °C in argon for 2 h 30 min as indicated in the Sample History section. Therefore, we can discard a thermal activation process for the motion of the twin boundary at 450 °C.

As a result, a chemical driving force, i.e., the difference of interface energies between {111} and {115} free surfaces exposed to CO can cause migration of the twin boundary in the direction of the interface with higher energy, i.e. Pt{111}/CO, and explain the twinning mechanism observed for conditions (2) and (9). Note that twinning is not observed for condition (8), while the particle is exposed to CO as in condition (2) (see Fig. 2). The oxidation of the Pt surface under O₂ rich conditions (condition (6)) followed by reduction by CO (condition (8)) will thus unlikely reestablish the initial surface. For instance, surface roughening has been observed after oxide reduction[63,64]. The different surface and strain states, which depend on the history of the crystal, may explain this absence of twinning.

In addition to the anisotropy of interfacial energy between the free surfaces exposed to CO, the change of local strain/displacement at the surface of the particle during reaction may also contribute to the twin boundary migration process. The interaction of gas species with the surface results in local strain gradients. The initial crystal shows a large compressive strain at its twin boundary. The resulting stress may favor the nucleation of dislocations allowing the boundary to move. An increase of the out-of-plane displacement at the particle top surface (see Fig. S6) was observed during detwinning. Hence, deformation-induced detwinning may occur. A striking result from the strain analysis is the remnant compressive out-of-plane strain observed at the position of the twin boundary formed in condition (2) during the detwinning process. Interestingly, a surface step can be observed at the location left by the migrating TB (see Figs. 2e, f and 4e, f). The remnant compressive strain at the surface location left by the

migrating TB may be explained by shear-coupled grain boundary motion[65]. Note that this strain increases from Fig. 4(b) to (f) and suggests that the TB leaves defects behind. However, since we do not reconstruct phase jumps typical of dislocations (see Fig. S6), we can infer that the Burgers vector ($\vec{b}$) of these defects does not fulfill the visibility condition[66]: $\vec{g} \cdot \vec{b} \neq 0$ (where $\vec{g}$ corresponds to the **111** diffraction vector; actually two out-of the three possible slip systems in the $(\bar{1}\bar{1}1)$ plane fulfill the invisibility conditions) or that their induced displacement field is below the spatial resolution of the measurements as they are closely spaced or that the dislocation movement related to twin boundary migration is too fast to be detected by our technique. At the end of the detwinning process, the TB does not migrate back to the initial configuration (i.e., condition (1)); it reaches a $[\bar{1}\bar{1}2]$ grain boundary. This GB was not present initially in the crystal at the position observed in conditions (6) and (8). It was probably in the crystal and could have migrated during twinning/detwinning due to the driving force associated with the changing strain distribution in the particle, with the anisotropy of the particle-substrate interface[56] or with thermal grooving[67,68]. Owing to the mask processing route, it was possible to locate the Pt particle (see SEM image displayed in Fig. 1a). We performed micro-Laue diffraction at the BM32-ESRF beamline after the BCDI experiment. It has been observed that the crystal contains at least three large grains with their [111] directions aligned close to the substrate normal (see Fig. S24); this demonstrates the complex microstructure of the crystal. It should be noted that low-angle GBs are sometimes observed in the particles produced by solid state dewetting, and these may migrate during prolonged heat treatments[69]. Deviations from the perfect [111] orientation in the neighboring grains may provide the driving force for the GB migration due to the energy anisotropy of the particle-substrate interface[70]. The elevated temperatures ensure sufficiently high GB mobility (associated with diffusional movement of Pt atoms in the GB core) enabling its migration. The mobility of the Pt atoms during reaction is not only evidenced by the GB migration but also by the faceting of the particle. Knowing the 3D orientation of the particle, it is possible to determine the ($hkl$) Miller index of the newly grown facets (see Fig. 2). During CO oxidation (Fig. 2d), several facets are well evidenced: (111), {102}- and {112}-type facets. Particle refaceting has been previously observed during CO exposure[71–74] or CO oxidation[75,76]. Surface diffusion, oxide formation, and CO adsorption may explain the surface change. As an example, Vendelbo et al.[4] demonstrated that the stabilization of the facets is linked to the chemisorption of oxygen on the facet sites in a low CO concentration regime. This restructuring phenomenon may have important implications for heterogeneous catalytic reactions.

In summary, we observed twin boundary migration (twinning/detwinning) during reaction depending on the gas feed composition and temperature. As demonstrated by DFT calculations, twinning can be correlated with the relative change in interfacial energy between the parent and twin-free surfaces exposed to CO. Facet formation and migration of the $[\bar{1}\bar{1}2]$ grain boundary is in line with high mobility and diffusivity of Pt atoms during reaction, either related to the gas interaction, temperature and/or strain. After twinning, a remnant compressive strain was observed at the position of the twin boundary suggesting that the twin boundary leaves trailing defects behind, and this may be explained by shear-coupled grain boundary motion. The non-invasive nature of Bragg coherent diffraction imaging is particularly well adapted to dynamically study the structure (strain, shape, faceting, defects) of nanoscale materials under external stimuli (i.e., pressure, temperature) in reactive environments at the individual particle level.

## Methods

**Sample preparation**. A 30 nm-thin platinum film was deposited by an electron beam evaporator on a lithographically patterned α-Al$_2$O$_3$ sapphire (c-axis oriented) substrate. The substrate was thoroughly cleaned with a sequence of detergent water, acetone, methanol, isopropanol, followed by immersion of 10 min in Piranha solution (H$_2$SO$_4$:H$_2$O$_2$ 2:1 by volume), and subsequently washed with DI water. A standard photolithography method was employed to prepare a patterned layer of photoresist on sapphire prior to the electron beam evaporation of Pt. The patterning was performed by photoresist spin coating, mask exposure, and development. After the Pt film deposition at room temperature, the lift off process in hot N-Methyl-2-pyrrolidone and acetone was performed. The thin film was subsequently dewetted at 800 °C in ambient air for 24 h in order to form single crystalline Pt particles of different sizes (100–500 nm in lateral size). The low (800 °C) annealing temperature is prone to the formation of twin and grain boundaries. The lithographic/mask processing route ensured that some of the crystallites are isolated. The obtained crystallites exhibited the same well-defined out-of-plane orientation as the original thin Pt film, with the Pt [111] direction being normal to the (0001) sapphire surface.

**Gas reactor**. We used a furnace compatible with nano X-ray beam and developed for in situ gas experiments[33]. The temperature was measured using a S-type thermocouple. The thermocouple was located in the button heater of the reactor. A gas flow of 50ml/min was delivered with a gas panel developed at TU Eindhoven. The gas composition inside the catalytic chamber was measured using a gas spectrometer (Pfeiffer vacuum ThermoStar GSD320T). The total pressure of the gas reactor was close to atmospheric pressure.

**Sample history before BCDI measurements**. Prior to the BCDI measurements shown in this paper, the sample has been exposed to several gas mixtures, including (1) H$_2$ (20% in Ar), (2) pure Ar, (3) CO (2% in Ar), (4) CO (2%) and O$_2$ (4%) in Ar, and finally (5) O$_2$ (4% in Ar) at 300 °C. Then, O$_2$ was replaced by pure Ar during sample re-alignment at 450 °C. The first BCDI measurement at 450 °C happened 2h30 after introducing pure Ar.

**BCDI measurements**. BCDI measurements were performed at the upgraded ID01 beamline of the ESRF synchrotron[77]. A coherent portion of the beam was selected with high precision slits by matching their horizontal and vertical gaps with the transverse coherence lengths of the beamline: 200 μm (vertically) and 60 μm (horizontally). The required beam size was obtained with a Kirkpatrick-Baez (KB) mirror, which focused the beam down to about 500 nm (horizontal) × 680 nm (vertical) full width at half maximum, as determined by knife-edge scans. This beam size ensured the full illumination of a single nanoparticle. An order sorting aperture was placed after the KB mirror to remove any parasitic scattering. The intensity distribution around the **111** Pt reflection was measured in a vertical coplanar diffraction geometry. The sample was rotated around the Bragg condition by steps of 0.01° over a total range of 1° in order to sample the **111** Pt Bragg peak. The BCDI experiment was performed at a beam energy of 9 keV (wavelength of 1.3775 Å) using a Si (111) monochromator. The diffracted beam was recorded with a 2D Eiger2M photon-counting detector (2164 (vertical) x 1030 (horizontal) pixels, each pixel 75 x 75 μm) positioned on the detector arm at a distance of 0.86 m. The detector distance as well as the rocking angle increment was chosen in order to ensure the oversampling of interference fringes. A typical counting time was 1 s per angle without beam attenuation. The sample was mounted on a Physik Instrumente Mars xyz piezoelectric stage with a lateral stroke of 100 μm and a resolution of 2 nm, sitting on a hexapod that was mounted on a (3 + 2 circles) goniometer.

**Phase retrieval**. Phase retrieval was carried out on the raw diffracted intensity data using PyNX package, imposing at each iteration that the calculated Fourier intensity of the guessed object agrees with the measured data. The metric used to estimate the goodness of fit during phasing was the free log-likelihood[78], available in PyNX. Defective pixels for experimental data and gaps in the detector were not used for imposing the reciprocal space constraint mentioned above and thus were evolving freely during phasing. The initial support, which is the constraint in real space, was estimated from the autocorrelation of the diffraction intensity. A series of 1400 Relaxed Averaged Alternating Reflections (RAAR)[79] plus 200 Error-Reduction steps[80,81], including shrink wrap algorithm[82] every 20 iterations were used. The phasing process included a partial coherence algorithm to account for the partially incoherent incoming wave front[83]. After removing the phase ramp and phase offset, the data was finally interpolated onto an orthogonal grid for easier visualization. To ensure the best reconstruction possible, we kept only the best 10 reconstructions (with lowest free log-likelihood) from 1000 with random phase start and performed the decomposition in modes[78].

The spatial resolution has been determined by the normalized phase retrieval transfer function (PRTF)[84] at a cutoff value of 1/e.

The nanocrystal volume was estimated by summing all the voxels of the reconstruction having a Bragg electron density larger than 23% of the maximum of the Bragg electron density. The cutoff is chosen from the histogram of the recovered modulus[39]. The sum has been then multiplied by the voxel size. The

error bars represent a range of ±5% of the measured volume using the 23% isosurface.

**Postmortem analysis**. Scanning electron microscopy (SEM) measurements have been performed ex situ after the BCDI experiment. It was possible to locate the measured nanoparticle (see Fig. 1a). Electron back-scattered diffraction (EBSD) and micro-Laue diffraction have been applied on the nanoparticle to determine the orientation of the different sub-grains composing it. The micro-Laue diffraction experiment has been performed at the BM32-ESRF beamline. The white beam has been focused down to 400 x 400 nm using KB mirrors.

**Density functional theory**. The density functional theory (DFT) calculations were performed by employing the PWSCF utility of Quantum ESPRESSO[85]. Ultrasoft pseudopotentials with non-linear core-correction were employed[86]. Energy cutoffs of 55 Ry and 550 Ry were used respectively for representation of the wavefunctions and charge density.

**Free-surface energies**. Free-surface energies, $\gamma_{hkl}^{vac}$, are computed using Eq. 1:

$$\gamma_{hkl}^{vac} = \frac{E_{slab} - NE_{bulk}}{2A_{slab}} \qquad (1)$$

where $N$ is the number of Pt atoms in slab and $A_{slab}$ is the area of the slab. For the bulk, a full geometrical optimization of the one-atom primitive cell of Pt (fcc) is carried out using an 8×8×8 k-point mesh to sample the integration over the Brillouin zone. The slab is built out-of 10 atomic layers with a vacuum of 18 Å. Three outer layers of Pt atoms on each side of the slab are allowed to geometrically relax for the calculation of the total energy and convergence is achieved when all forces on the loose Pt atoms are lower than 0.001 Ry/Bohr. The k-points mesh for the slabs is chosen by rescaling the mesh of the primitive cell according to the relative sizes. The surface energies computed for {111} and {511} surfaces using different exchange and correlation functionals (XCF) are reported in Table S2, together with results from previous studies, for comparison. Some XCFs have been used in conjunction with Grimme's DFT-D3 van der Waals' correction without damping as implemented in PWSCF[87]. The unit cell employed for the {511} slab is shown in Figure S7 and is characterized by five surface atoms.

**Gibbs free energy of adsorption**. To study the stability of the different adsorption configurations as a function of gas pressure and temperature, we compute the Gibbs free energy of adsorption as follows:

$$\Delta G_{ads}^{CO}(P, T) = \frac{G_{CO@slab} - G_{slab} - n\mu_{CO}(p, T)}{n} \qquad (2)$$

The chemical potential of CO is determined by the thermodynamic equilibrium with the gas phase reservoir and it is computed as follows:

$$\mu_{CO}(p, T) = E^{elec} + E^{ZPE} + \tilde{\mu}_{CO}(T, p^0) + k_B T \ln\left(\frac{p_{CO}}{p^0}\right) \qquad (3)$$

the first two terms are the electronic and zero-point energies and are computed using DFT. The term $\tilde{\mu}_{CO}(T, p^0)$ includes thermal effects contribution from vibrations and rotations of the molecules together with the ideal gas entropy at $p^0 = 1$ atm. In this work, we take the experimental values from the JANAF thermochemical tables[88]. Specifically, we take $\tilde{\mu}_{CO}(T, 1)$ equal to −0.53 eV at 300 K and −1.40 eV at 700 K. For the free slab and the CO adsorbed configurations, we approximate G by E, i.e., the total energy (see next section).

**Interfacial energy**. Interfacial energy of the Pt/CO interface as a function of CO coverage is defined in Eq. 4[76,89]; where $A_{slab}$ is the unit area of the slab, $n$ the number of CO molecules adsorbed on the Pt surface of the slab, and $E_{ads}$ the adsorption energy (Eq. 5).

$$\gamma_{hkl}^{CO}(n) = \gamma_{hkl}^{vac} - \frac{nE_{ads}(n)}{A_{slab}} \qquad (4)$$

$$E_{ads}(n) = \frac{E(nCO/Pt) - nE(CO) - E(Pt)}{n} \qquad (5)$$

We note that the adsorption energy, rather than the Gibbs free energy of adsorption, is employed in Eq. (4) for consistency with the calculation of $\gamma_{hkl}^{vac}$ which is computed using total energy differences. For the calculation of the interfacial energy, the energetically preferred configuration of the adsorbed CO molecules at each coverage is used. Adsorption energies are computed by relaxing the CO molecule and the first three Pt layers. To model adsorption of CO on Pt {111}, a 4 × 4 supercell of a slab of six layers of Pt atoms with CO molecules adsorbing only on one side of the slab is considered. The configurations of adsorbed CO on {111} are taken from an earlier work[53]. For the {511} surface, a Pt slab of seven atomic layers is used. Several low- and high-coverage configurations are computed (see Figure S7 and Figures S11-S22) and the lowest-energy configuration at a given coverage is used to compute $\gamma_{511}^{CO}$ (see Table S2). Due to the asymmetry of the slab upon CO adsorption, the CO/Pt calculations are performed by employing a dipole-correction method[90] in order to cancel the unphysical dipole interactions between

periodic images. The $\gamma_{111}^{CO}$ and $\gamma_{511}^{CO}$ values computed using different XCF at varying coverages are reported in Table S3. These values are then rigidly shifted by a constant for representation in Figure S9 and Fig. 5. This constant is determined by mapping the $\gamma_{hkl}^{vac}$ to the experimental value (see Table S2) and is applied also to shift $\gamma_{511}^{CO}$. This rigid-shift strategy allows to report non-negative values of $\gamma_{hkl}^{CO}$, while maintaining the DFT-predicted relative difference between the {111} and {511} curves.

## Data availability

The data reported in this paper is available upon request.

## Code availability

The phasing algorithm PyNX[91] is available at http://ftp.esrf.fr/pub/scisoft/PyNX/. The scripts used for BCDI data post-processing, PRTF calculations and stereographic projections belong to the BCDI[92] package (https://doi.org/10.5281/zenodo.3257616), that can be downloaded from PyPI (https://pypi.org/project/bcdi/) or GitHub (https://github.com/carnisj/bcdi).

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

## Acknowledgements

The authors are grateful to the European Synchrotron Radiation Facility for allocating beamtime. BCDI measurements were performed at beamline ID01. We thank Hamid Djazouli for his excellent support during the preparation of the experiment. Laue microdiffraction measurements were measured at beamline BM32. The authors would like to thank Vincent Favre-Nicolin for his continuous improvement of the PyNX package used here for phase retrieval. E.J.M.H. acknowledges financial support of NWO TOP and KNAW CEP grants. M.-I.R. acknowledges financial support from ANR Charline (ANR-16-CE07-0028-01) and ANR Tremplin ERC (ANR-18-ERC1-0010-01). This project has received funding from the European Research Council (ERC) under the European Union's Horizon 2020 research and innovation programme (grant agreement No. 818823). N.G., E.A. and E.R. wish to thank the support by a grant from the Ministry of Science & Technology, Israel & France's "Centre National de la Recherche Scientifique (CNRS)". The thin film processing was performed at the Micro-Nano Fabrication and Printing Unit (MNF&PU), Technion. The calculations presented in this paper were performed using the Froggy platform of the CIMENT infrastructure (https://ciment.ujf-grenoble.fr), which is supported by the Rhône-Alpes region (GRANT CPER07_13 CIRA) and the Equip@Meso project (ANR-10-EQPX-29-01) of the "Programme Investissements d'Avenir" supervised by the "Agence Nationale pour la Recherche". This work was supported by the Netherlands Center for Multiscale Catalytic Energy Conversion (MCEC), an NWO Gravitation programme funded by the Ministry of Education, Culture and Science of the government of the Netherlands.

## Author contributions

J.C., L.G., F.O., J.P.H., S. Labat, J.-S.M. and M.-I.R. carried out the experiment. M.I-R directed the project. J.C., M.-I.R., S. Labat analysed the data. J.C., M.-I.R, S. Labat, L.F., M.T., L.W., M.D., S.J.L., E.H., T.S. and O.T. interpreted the data. A.C. and L.F. performed electron microscopy measurements. A.K. and R.P. carried out DFT simulations. N.G., E.A. and E.R. prepared the samples. J.C. and M.-I.R. wrote the manuscript. All authors reviewed, discussed the manuscript and have given approval to its final version. The authors declare no competing interests.

## Competing interests

The authors declare no competing interests.
