## [Peer Review File · Nature Communications]

REVIEWER COMMENTS

Reviewer #1 (Remarks to the Author):

Report on manuscript entitled „Twinning/detwinning in an individual platinum nanocrystal during catalytic CO oxidation“ by Carnis et al.

The manuscript discusses the structural evolution of a twinned Pt particle with a size of 800 nm under gas exposure to mixtures of oxygen and CO with various ratios at elevated temperatures investigated by Bragg coherent diffraction imaging. The experiment seems to be carefully conducted and the x-ray data analysis is thorough. The main finding is that the termination on one side of the part of the particle which is imaged, is changing position during the experiment. This is as such an interesting observation, and the interpretation of the authors might be correct, that it is a moving twin grain boundary. For an 800 nm sized particle, I however doubt that the observation is relevant for heterogeneous catalysis (typical particle size 1-20 nm), which is the main motivation of this study. One very problematic point of this study is, that only one part of the particle is imaged, because only one Bragg reflection was investigated. So, it is very unclear, how the rest of the particle is evolving during the process observed, which makes the interpretation of the results rather difficult. The authors state that their “Density-functional theory calculations show that twinning can be correlated with the relative change in the interfacial energies of the free surfaces exposed to CO”. I disagree with this conclusion, since the resulting structure under CO flow conditions at 450°C (2) and (8) is not identical. In addition, their conclusion is based on the argument, that the twinned part of the particle exposes a 511 surface to the gas atmosphere and that a 511 surface exhibits a higher adsorption energy for CO as compared to 111 which the authors argue is the driving force for the moving twin boundaries. I do not support these conclusions for the following arguments:

- The twinned part of the particle is not imaged, therefore the authors do not know what the surface morphology and termination of this part is
- If the increase in 511 surface area would be the driving force, the whole imaged part of the particle would be expected to transform, which obviously does not take place
- The calculated differences in CO adsorption energies between the 111 and the 511 surface are small and change sign as a function of coverage
- The coverage under the experimental conditions is not well enough known, the calculations neglect the chemical potential of the CO gas phase. The relevant quantity is the Gibbs free energy per surface area. The values given in Ref. 49 are too inaccurate and depend on the functional used.
- It is unclear, if the energy gain from CO on 511 facets is high enough to move a twin boundary extending through the whole particle. The energy should be compared with the energy needed to move the twin boundary.

In more detail, I have the following comments:

Page 2, last paragraph: the total pressure in the cell is not given

Page 3, first paragraph: how was it determined that there is a {111} twin boundary?

Page 3: it is unclear, how the lattice of the twin is oriented in 3D with respect to the parent grain lattice

Page 3: last paragraph: what does it mean “data not saved” ?

Page 4, top: why was a value of 23% chosen as a cutoff for the density in Fig. 2?

Page 4, middle: after CO introduction, the volume decreases. What is the experimental evidence that this is due to twinning?

Page 4, last paragraph: under condition (8), CO, a different grain boundary is formed. What is the exact orientation of the grain boundary? It is not exactly perpendicular to the 111 planes, when looking at Fig. 2f, so [-1,-1,2] does not match.

Page 5, top: what is the explanation for the formation of the specific small side facets under conditions 4 and 6?

Page 5, discussion: the twin boundary energy value for Pt is missing.

Page 6, bottom: the energy balance between twin boundary energy and surface energy gain for CO on Pt 511 versus 111 is missing. It is unclear, if the energy gain at all is high enough for twin movement. It is unclear, why twin movement at 500°C can also be thermally activated and at 450°C not. This is not such a big temperature difference.

Page 7, top: References 53 and 54 refer to Pt 110 surface oxide reduction induced roughening at lower temperatures. This situation can not be compared with the present situation.

Page 7: it is unclear from where a [-1,-1,2] grain boundary could come from and why it should be more stable than the {111} grain boundary.

Page 8: in the conclusions it is stated: "Facet formation and migration of the [-1,-1,2] grain boundary are in line with high mobility and diffusivity of Pt atoms during reaction." I disagree with this statement: both have not the same origin: the first one is due to the interaction with the gas, the second one because of temperature and / or strain.

Page 9: sample preparation: the Pt deposition temperature is not given

Gas reactor: the total pressure in the reactor is not given. Mass spectrometer data are missing, showing the CO oxidation activity of the system

Page 11: DFT: the CO chemical potential is not included in the calculations

Page 18: Fig 3: what is the meaning of the dashed circles?

SI, page 2, Fig. 1: what do A-F correspond to?

Reviewer #2 (Remarks to the Author):

Twinning is found in a Pt sub-micro crystal investigated in an in-situ reactive gas environment using Bragg coherent diffraction imaging. Twinning is uncommon in Pt bulk but does occur in nanocrystals. Half of a twin is observed here terminating with a large 111 facet identified from the crystal morphology, measured at a different 111 Bragg peak. The other half of the twin with opposite stacking sequence is not observed, although several other BCDI publications have been able to see more than one Bragg peak from the same crystal. The work of F. Hofmann (at least) should be referenced to admit this shortcoming.

Nevertheless, the authors were able to document clear changes of crystal volume during the catalytic reactions, associated with forth-and-back motion of the stacking fault. This is associated with a restacking of the {111} planes that brings more (or less) of the crystal into diffraction. Such motion of twin boundaries is expected under the influence of external forces as probably occurs here. The uniqueness of the work is the chemical origin of these restacking forces.

As mentioned, there should be dislocation migration associated with the restacking. The authors state that "However, since we do not reconstruct phase jumps typical of dislocations". I think this is a misunderstanding of the expected role of dislocations in the evolution of the crystal: to explain restacking behaviour, as seen here, dislocations are expected to traverse the twin boundary plane, not be embedded inside the half-crystal seen. The arrow in Fig 2e points at a possible dislocation crossing the internal boundary. The manuscript should be revised to correct this discussion.

It is stated repeatedly that "twinning was observed" when this is factually incorrect. The crystal contains a twin which migrates forth and back during the experiment. No formation or disappearance of twins has been seen during the experiment. The manuscript needs to be reworded to say this. The

title needs revising too, because it is highly misleading to imply the twinning is occurring.

Please remove unconventional boldface notation "specular $\{111\}$ Pt Bragg reflection" etc

Reviewer #3 (Remarks to the Author):

In this paper, the structure of single ~ 200 nm Pt particles is followed during exposure to CO and O₂ at 450 and 500 C using Bragg coherent diffraction imaging, a relatively new X-ray diffraction technique. The measured change in structure during the experiment is supported by some DFT calculations.

The dynamic behavior of Pt catalysts under CO oxidation conditions is well known and has been studied with TEM (e.g., Vendelbo et al., Nat. Mater. 2014), STM (e.g., Tao et al., Science 2010) and IR (e.g., Avanesian et al., JACS 2017). This work is a nice addition to these earlier studies. I would recommend citing some of this complementary work.

The limitation of the study to a single, large Pt particle raises the question of the statistical relevance of the observed twinning/detwinning. In situ TEM is also limited to a few clusters, but at least statistical averaging is typically done. 200 nm is also very large for a catalytic Pt particle. These limitations should be clearly discussed and data points for a few particles should be included if possible, maybe only at initial and final conditions.

Fig S3 suggest that detwinning is mostly a function of time, and not conditions. This seems logical considering the high energy of the twinning defect. Is CO adsorption driving twinning/detwinning as suggested by the DFT calculations? Or is the local heat of the CO oxidation reaction relevant? Is a control experiment possible?

Was activity measured during the experiment? It would be nice to complement the structural data.

The particle preparation procedure seems relevant – during preparation the particles were exposed to air and 800 C. How does this affect the resulting particles? How common are twinning defects in these particles?

The experiments show both strain and twinning/detwinning. The calculations only consider adsorption on the (511) and the (111) surface. Strain effects were not considered, though they are at least as important. Twinning is a bulk defect. How is this connected to surface calculations?

The surface energy (γ) is a free energy. The adsorption entropy has an important effect on the surface coverage and the stability and therefore needs to be included in the calculations.

Why were only the (511) and (111) surface considered? The (100) surface is also highly stabilized by CO adsorption, as shown in earlier work.

Title: Twin boundary migration in an individual platinum nanocrystal during catalytic CO oxidation

No. NCOMMS-21-02033-T

ANSWERS TO REFEREES

We submit a revised manuscript and a supporting document for Reviewers with all the corrections highlighted in red. All responses to Reviewers and corrections to the manuscript have been listed as follows one by one.

Reviewer #1:

The manuscript discusses the structural evolution of a twinned Pt particle with a size of 800 nm under gas exposure to mixtures of oxygen and CO with various ratios at elevated temperatures investigated by Bragg coherent diffraction imaging. The experiment seems to be carefully conducted and the x-ray data analysis is thorough. The main finding is that the termination on one side of the part of the particle which is imaged, is changing position during the experiment. This is as such an interesting observation, and the interpretation of the authors might be correct, that it is a moving twin grain boundary.

Answer 1: We thank the reviewer for her/his positive evaluation.

For an 800 nm sized particle, I however doubt that the observation is relevant for heterogeneous catalysis (typical particle size 1-20 nm), which is the main motivation of this study.

Answer 2: We agree that our particle is large for a catalytic Pt particle. The studied particle can be considered as a model particle. It can be observed that even large particles (see the work of Abuin *et al.*, ACS Appl. Nano Mater. 2019, 2, 4818–4824 (2019)) show strain or shape evolution during reaction. We see it as a bridge between large (10 nm) single crystal facets (*e.g.* Pt(111)) and catalytic nanoparticles.

One very problematic point of this study is, that only one part of the particle is imaged, because only one Bragg reflection was investigated. So, it is very unclear, how the rest of the particle is evolving during the process observed, which makes the interpretation of the results rather difficult.

Answer 3: It is possible to measure several Bragg peaks of the same crystal under ideal conditions but this remains challenging even in *ex situ* experiments. Therefore, only one part of the particle (the parent grain) has been measured. But from the structural evolution of the parent grain, we can deduce grain boundary motion. We would like to note in this respect that tracking the Bragg reflection of one of the two grains abutting the grain boundary (somewhat reduced in intensity because of the beam positioning at the grain boundary) has been widely employed in the studies of kinetics of grain boundaries (see D.A. Molodov *et al.*, Acta mater. 46 (1998) 553).

The authors state that their “Density-functional theory calculations show that twinning can be correlated with the relative change in the interfacial energies of the free surfaces exposed to CO”. I disagree with this conclusion, since the resulting structure under CO flow conditions at 450°C (2) and (8) is not identical.

Answer 4: The resulting structure under CO flow conditions at 450°C for the identical conditions (2) and (8) is not the same. We think that asymmetry in CO adsorption energy initiated the migration of the twin boundary, but the full story may be more complex especially because we have also O₂ adsorption between conditions (2) and (8). In condition (6), we expect O₂ to remove progressively the adsorbed CO. Once CO is reintroduced in (8), a [-1-12] large angle grain boundary is observed, which may be more stable than a [-1-11] twin boundary. But then it goes back to a [-1-11] at 500°C. It is a complex interplay between temperature, adsorption and stress.

In addition, their conclusion is based on the argument, that the twinned part of the particle exposes a 511 surface to the gas atmosphere and that a 511 surface exhibits a higher adsorption energy for CO as compared to 111 which the authors argue is the driving force for the moving twin boundaries.

Answer 5: This is one of the arguments that the twin boundary movement is initiated by chemical changes which induce surface energies changes. Mechanical changes (stress) as well as temperature are also arguments in favor of the twin boundary movement as explained in the conclusion.

- The twinned part of the particle is not imaged, therefore the authors do not know what the surface morphology and termination of this part is

Answer 6: The twinned part of the particle has not been imaged. But from the orientation of the grain boundary and from the change of volume of the particle, we can infer that the boundary is a {111} twin boundary, which implies that the twinned part is [115] oriented. We have included a new figure in Supplementary Materials (Fig. S1) illustrating the atomistic configuration of the parent and twin grains. If the not imaged part was from misoriented [111]-oriented grains we should have measured them as we rocked the sample over at least 2° with Bragg coherent diffraction imaging, which we did not.

- If the increase in 511 surface area would be the driving force, the whole imaged part of the particle would be expected to transform, which obviously does not take place

Answer 7: We think that the {511} surface area is the driving force. The process takes time. As shown in Figure S3, detwinning evolves linearly with time. The whole imaged part did not completely transform and that is why the twin migrates back during the experiment.

- The calculated differences in CO adsorption energies between the 111 and the 511 surface are small and change sign as a function of coverage

Answer 8: We agree that the calculated difference in CO adsorption energies is relatively small but this small variation may initiate the movement of the twin. The particle was annealed in air at 800°C for 24 hours. Before the BCDI measurements presented in this manuscript, the sample was already at 450°C in argon for 2h 30 min as indicated in the Sample History section. Therefore, we can discard a thermal activation process for the motion of the twin boundary at 450°C. A good example of small difference of surface energies driving the grain boundary migration is abnormal grain growth in thin films. Indeed, Humphreys and Hatherly have shown that only 2% difference in surface energies of two neighboring grains can drive the abnormal growth (see F.J. Humphreys, M. Hatherly, Recrystallization and Related Annealing Phenomena, 2nd Edition, Elsevier, p. 378 (1995)). This example is now mentioned in the revised version of the manuscript.

- The coverage under the experimental conditions is not well enough known, the calculations neglect the chemical potential of the CO gas phase. The relevant quantity is the Gibbs free energy per surface area. The values given in Ref. 49 are to inaccurate and depend on the functional used.

Answer 9: According to Ref. [<https://pubs.acs.org/doi/10.1021/acscatal.8b02371>] the coverage that should be expected during our experiment (at 450 degrees and 2% CO) is between 0.5 and 0.6 ML for a (111) surface. Since the adsorption energy is always larger for (511) compared to (111) we can expect a higher coverage for the former at fixed temperature and, therefore, a non-negligible difference of the corresponding gamma values. The Gibbs free energy is the relevant quantity to study adsorption properties and in this case, we agree that the chemical potential of CO in gas phase should be accounted for. In order to compare the thermodynamic stability of solid-gas interfaces the relative quantity is the interface free energy. We approximate free energies using total energies (thus neglecting thermal contributions) as often done in the literature. This is obviously motivated by the substantially heavier computational cost imposed by the calculations of the vibrational properties of such large supercells. This

approximation is however justified by findings reported in Refs. <https://pubs.acs.org/doi/10.1021/acscatal.9b01840> and <https://pubs.acs.org/doi/10.1021/acscatal.8b02371>. The former employs DFT calculations to show that differences in adsorption energies of CO on Pt surfaces (computed using total energy differences) are very similar (within the kJ/mol) to differences in the Gibbs free energy of adsorption, when different surfaces are considered.

For example, we report the adsorption energy (binding energy), heat of adsorption (enthalpy) and Gibbs free energy for the most stable configuration of CO adsorbed onto different Pt free surfaces, in kJ/mol.

Pt(111): -140, -140, -68

Pt(110): -167, -170, -96

Pt(100): -159, -164, -89

The second article cited above shows the same result when comparing differences (among different coverages) in adsorption energies and Gibbs free energies of CO adsorbed onto Pt(111).

Regarding free-surfaces, we expect that the dominant contribution to the free energy comes from the cleavage of the surfaces and bond breaking (see Ref. <https://www.pnas.org/content/114/44/E9188>). We have added a sentence in the Method section to clarify this:

We note that the adsorption energy, rather than the Gibbs free energy of adsorption, is employed in equation (2). However, because differences in the Gibbs free energy of adsorption of CO computed for different free surfaces have been shown to be almost equal to the differences in the adsorption energies computed using equation (3) (Ref. <https://pubs.acs.org/doi/10.1021/acscatal.9b01840>), we are confident that our approach provides a good estimate of the difference between $\gamma_{111}^{CO}(n)$ and $\gamma_{511}^{CO}(n)$.

Regarding Ref. 49, only one functional is used and we do not understand the comment about inaccuracy since the (111) surface is not studied in that work. The dependence of the adsorption energy on the used functional is a well-known fact in the literature (we do cite several works on this respect) and we do not hide this issue from the reader. On the contrary, we show the dependence of both the free surface energies and adsorption energies on the functional demonstrating that the (511) is always preferred regardless of the functional used. For all these reasons we believe that our computational scheme, employed in many recent studies (<https://pubs.acs.org/doi/abs/10.1021/jacs.7b01081>) is sound and our conclusion robust.

Finally, we would like to note that the (511) surface is more disordered than its densely packed (111) counterpart and, hence, in most likelihood it also exhibits higher entropy. This should make the difference in free energies between the two surfaces even larger than the difference of energies.

• It is unclear, if the energy gain from CO on 511 facets is high enough to move a twin boundary extending through the whole particle. The energy should be compared with the energy needed to move the twin boundary.

Answer 10: For a CO coverage of 0.667 ML, a 300mJ/m² difference in surface energy is observed. We expect a lower CO coverage (0.5 – 0.6 ML, corresponding to 0.073 CO/Å² and 0.088 CO/Å²) for which a difference energy up to about 200mJ/m² is observed.

The driving forces associated with the polycrystalline structure evolution are generally small, much smaller than, for example, the heat of fusion. Let us estimate the typical driving force associated with grain boundary migration in Pt. It was shown that the recrystallization in heavily cold rolled Pt foils begins at temperatures as low as 400 °C (Yu. N. Loginov et al., Platinum Metals Rev. 51 (2007) 178). The driving force of recrystallization, E_{rec} , is mainly related to the energy of stored dislocations, and can be roughly estimated as (see F.J. Humphreys, M. Hatherly, Recrystallization and Related Annealing Phenomena, 2nd Edition, Elsevier, p. 17):

$$E_{rec} = 0.5\rho Gb^2 \quad (1)$$

The maximum dislocation density in heavily deformed metal with high stacking fault energy is of the order of $\rho \approx 10^{15} \text{ m}^{-2}$. For Pt, $b = 0.277 \text{ nm}$, and G can be estimated by its isotropic value of 62 GPa. Then, $E_{\text{rec}} \approx 2.4 \times 10^6 \text{ J/m}^3$. On the other hand, the driving force for twin boundary migration in the geometry of our work, E_s , is

$$E_s = \frac{\Delta\gamma}{h} \quad (2)$$

where $\Delta\gamma = 200 \text{ mJ/m}^2$ is the difference of the Pt/CO interface energies, and $h \approx 110 \text{ nm}$ is the particle height. Eq. (2) can be derived by considering infinitesimal lateral displacement of the boundary. With these data, $E_s \approx 1.8 \times 10^6 \text{ J/m}^3$. One can see that the two driving forces are of the same order of magnitude and, therefore, the energy gain from CO segregation on (511) facet represents a viable driving force for twin boundary migration. This part has been included in the revised version of the manuscript.

Page 2, last paragraph: the total pressure in the cell is not given

Answer 11: The total pressure of the cell was not measured. But the reactor outlet was open to atmosphere, so we can estimate the pressure to be close to ambient pressure.

Page 3, first paragraph: how was it determined that there is a {111} twin boundary?

Answer 12: We assume that we have a {111} twin boundary as:

- we measure a {111} grain boundary using Bragg coherent diffraction
- we observed a change of the volume of the particle that we can associate to the movement of a {111} twin boundary.

Page 3: it is unclear, how the lattice of the twin is oriented in 3D with respect to the parent grain lattice

Answer 13: The parent grain has a [111] out-of-plane orientation, whereas the most probable twin has a [511] out-of-plane orientation (L. Shaw, et al., *Materials Science and Engineering: A* **480**, 75–83 (2008).). This reference has been included in the revised version of the manuscript.

Page 3: last paragraph: what does it mean "data not saved"?

Answer 14: The data were unfortunately not saved, meaning that the saving of the detector was turned off. We did not collect images for these conditions. We think that it is worth to mention it in the manuscript, as the particle underwent these conditions even if we have not saved images.

Page 4, top: why was a value of 23% chosen as a cutoff for the density in Fig. 2?

Answer 15: The cut-off is chosen from the histogram of the recovered modulus, see for example: J. Carnis *et al.*, *Scientific Reports* 9:17357 (2019). This is now mentioned in the new version of the manuscript.

Page 4, middle: after CO introduction, the volume decreases. What is the experimental evidence that this is due to twinning?

Answer 16: The decrease of the volume indicates a structural change of the particle. This decrease in combination with the migration of a {111} grain boundary is in line with twinning.

Page 4, last paragraph: under condition (8), CO, a different grain boundary is formed. What is the exact orientation of the grain boundary? It is not exactly perpendicular to the 111 planes, when looking at Fig. 2f, so [-1,-1,2] does not match.

Answer 17: In Fig. 2f left, the grain boundary is hidden by a small protuberance on the side of the crystal. On average, the new grain boundary is however perpendicular to (111) and [-1,-1,2] does match as demonstrated by the pole figures (Fig 3c).

Page 5, top: what is the explanation for the formation of the specific small side facets under conditions 4 and 6?

Answer 18: We observed refaceting during stoichiometric CO oxidation. Faceting is still observed during pure oxygen condition. Particle refaceting has been already observed during CO oxidation [Vendelbo *et al.*, Nat. Mater. 2014]. They have shown that at low CO pressure ($p_{CO} < 1$ mbar), both facets and step sites have significantly oxygen coverage, which tends to stabilize the facets. At higher CO pressure ($p_{CO} > 1$ mbar), the steps are stabilized compared to the facets because CO adsorption is stronger on steps than on the facets. In our case, we observe faceting during CO oxidation (decrease of the CO partial pressure), which is consistent with this model. This is now mentioned in the revised version of the manuscript.

Page 5, discussion: the twin boundary energy value for Pt is missing.

Answer 19: We now discuss the twin boundary energy value for Pt. Compared to other metals, the twin boundary energy for Pt is high (322 mJ/m^2) [N. Bernstein *et al.*, Phys. Rev. B **69**, 094116 (2004)].

Page 6, bottom: the energy balance between twin boundary energy and surface energy gain for CO on Pt 511 versus 111 is missing. It is unclear, if the energy gain at all is high enough for twin movement. It is unclear, why twin movement at 500°C can also be thermally activated and at 450°C not. This is not such a big temperature difference.

Answer 20:

We think that a direct comparison between the twin boundary energy and the energy gain due to CO adsorption (*i.e.* the driving force for twin migration) is not justified. The twin boundary has formed during sample fabrication (solid state dewetting) and its energy is unrelated to the driving force for migration during CO exposure. A more relevant parameter would be the mobility of the twin boundary and related with it energy barrier for twin boundary migration. The latter is probably related to the formation energy of the double kink on the twinning dislocation, which is quite low. However, estimating the twin boundary mobility even knowing the relevant activation energy is very difficult because of uncertainties in pre-exponential factor. In our answer to question 10 we analyzed the problem based on the literature data on recrystallization of cold rolled Pt foils. We estimated the driving force for recrystallization based on the maximum energy stored in dislocations, and found out that it has the same order of magnitude as the driving force for twin migration due to the difference of (511) and (111) facet energies. Since the cold rolled foils readily recrystallize at the temperature of 400 °C, the chemical driving force determined in our work should be capable of driving the twin boundary migration at the temperature of 450 °C.

We do not expect the twin movement to be thermally activated at 450°C due to the history of the sample. Under “thermally activated” we meant that the twin boundary migration can be due to any other driving forces for twin boundary migration different from the chemical driving force, such as residual stresses in the particle, or the anisotropy of the Pt-sapphire interface. Before the BCDI measurements presented in this manuscript, the sample was already at 450°C in Argon for 2h 30 min as indicated in the Sample History section. Therefore, we can discard thermally activated motion of the twin boundary due to all non-chemical driving forces at 450 °C. We agree with the Reviewer that our use the term “thermally activated motion” was somewhat confusing, and we removed it in the revised manuscript.

Page 7, top: References 53 and 54 refer to Pt 110 surface oxide reduction induced roughening at lower temperatures. This situation can not be compared with the present situation.

Answer 21: In ref. 54, the possibility of a certain degree of roughness of the surface for Pt (111) surfaces is mentioned.

Page 7: it is unclear from where a [-1,-1,2] grain boundary could come from and why it should be more stable than the {111} grain boundary.

Answer 22: Experimentally, we observe this [-1,-1,2] grain boundary in conditions (6) (partially) and (8). This observation is also confirmed by the stereographic projections. Possibly, this grain boundary was initially in the crystal and migrated because of temperature and/or strain.

Page 8: in the conclusions it is stated: "Facet formation and migration of the [-1,-1,2] grain boundary are in line with high mobility and diffusivity of Pt atoms during reaction." I disagree with this statement: both have not the same origin: the first one is due to the interaction with the gas, the second one because of temperature and / or strain.

Answer 23: We modified the text by adding: "either related to the gas interaction, temperature and/or strain". Note that strain can also be induced by the binding of adsorbates (CO in this case).

Page 9: sample preparation: the Pt deposition temperature is not given. Gas reactor: the total pressure in the reactor is not given. Mass spectrometer data are missing, showing the CO oxidation activity of the system

Answer 24: The Pt film was deposited at room temperature. The total pressure is close to atmospheric pressure. These facts are now mentioned in the manuscript. Mass spectrometer data are now included (see Fig. 2).

Page 11: DFT: the CO chemical potential is not included in the calculations

Answer 25: Please see our response above (Answer 9).

Page 18: Fig 3: what is the meaning of the dashed circles?

Answer 26: The dashed circles indicate the location of the peaks. They are guide for the eye. This is now clarified in the legend.

SI, page 2, Fig. 1: what do A-F correspond to?

Answer 27: A-F correspond to the labelling of the different images and indicate incremental gas conditions.

Reviewer #2:

Twinning is found in a Pt sub-micro crystal investigated in an in-situ reactive gas environment using Bragg coherent diffraction imaging. Twinning is uncommon in Pt bulk but does occur in nanocrystals. Half of a twin is observed here terminating with a large 111 facet identified from the crystal morphology, measured at a different 111 Bragg peak. The other half of the twin with opposite stacking sequence is not observed,

although several other BCDI publications have been able to see more than one Bragg peak from the same crystal. The work of F. Hofmann (at least) should be referenced to admit this shortcoming.

Answer A: A BCDI paper from A. Ulvestad dealing on twinned crystals is mentioned in the manuscript [Ulvestad, A., Clark, J. N., Harder, R., Robinson, I. K. & Shpyrko, O. G. 3D Imaging of Twin Domain Defects in Gold Nanoparticles. *Nano Letters* 15, 4066–4070 (2015)]. It is possible to measure several Bragg peaks of the same crystal [F. Hofmann et al, *Phys. Rev. Mater.* 4 (2020) 013801 or our recent work: F. Lauraux et al., *Crystal* (2021), 11(3), 312] under ideal conditions but this remains challenging even in *ex situ* experiments. The work of F. Hofmann (multiple reflections) is now mentioned in the manuscript.

Nevertheless, the authors were able to document clear changes of crystal volume during the catalytic reactions, associated with forth-and-back motion of the stacking fault. This is associated with a restacking of the {111} planes that brings more (or less) of the crystal into diffraction. Such motion of twin boundaries is expected under the influence of external forces as probably occurs here. The uniqueness of the work is the chemical origin of these restacking forces.

Answer B: We thank the reviewer for her/his positive evaluation.

As mentioned, there should be dislocation migration associated with the restacking. The authors state that "However, since we do not reconstruct phase jumps typical of dislocations". I think this is a misunderstanding of the expected role of dislocations in the evolution of the crystal: to explain restacking behaviour, as seen here, dislocations are expected to traverse the twin boundary plane, not be embedded inside the half-crystal seen. The arrow in Fig 2e points at a possible dislocation crossing the internal boundary. The manuscript should be revised to correct this discussion.

Answer C: We agree that during restacking behavior, dislocations are expected to traverse the twin boundary plane. Here by writing "However, since we do not reconstruct phase jumps typical of dislocations", we wanted to explain that we did not observe dislocations with our spatial resolution but dislocations may exist and are expected to traverse the twin boundary plane. The arrow in Fig. 2e does not point to a possible dislocation but to a change of orientation of the interface from [-1-11] to [-1-12]. Also, the dislocation movement related to twinning may be too fast to be detected by our technique. It is also possible that the dislocations have a Burgers vector of $\frac{1}{2}[101]$ or $\frac{1}{2}[1-10]$ and thus fulfil the invisibility conditions ($\mathbf{g} \cdot \mathbf{b} = 0$, with $\mathbf{g} = \mathbf{111}$). It is far from unlikely as two of the three possible Burgers vectors in this glide plane fulfill the invisibility conditions. This discussion has been included in the revised version of the manuscript.

It is stated repeatedly that "twinning was observed" when this is factually incorrect. The crystal contains a twin which migrates forth and back during the experiment. No formation or disappearance of twins has been seen during the experiment. The manuscript need to be reworded to say this. The title needs revising too, because it is highly misleading to imply the twinning is occurring.

Answer D: As the referee wrote, the crystal contains a twin which migrates forth and back during the experiment. We revisited the title and the manuscript to mention twin boundary migration instead of twinning/detwinning. But for some cases, we kept the words twinning and detwinning to clearly indicate in which direction (back or forth) the twin is moving.

Please remove unconventional boldface notation "specular $\backslash\mathbf{bf}\{111\}$ Pt Bragg reflection" etc

Answer E: In crystallography, the boldface notation indicates a reciprocal lattice point. As $\mathbf{111}$ Pt Bragg reflection refers to a lattice point, we prefer to keep it written in bold.

Reviewer #3:

In this paper, the structure of single ~200 nm Pt particles is followed during exposure to CO and O₂ at 450 and 500 C using Bragg coherent diffraction imaging, a relatively new X-ray diffraction technique. The measured change in structure during the experiment is supported by some DFT calculations. The dynamic behavior of Pt catalysts under CO oxidation conditions is well known and has been studied with TEM (e.g., Vendelbo *et al.*, *Nat. Mater.* 2014), STM (e.g., Tao *et al.*, *Science* 2010) and IR (e.g., Avanesian *et al.*, *JACS* 2017). This work is a nice addition to these earlier studies. I would recommend citing some of this complementary work.

Answer I: The works of Tao *et al.* and Avanesian *et al.* were already mentioned in the manuscript. We added a reference to the work of Vendelbo *et al.*

The limitation of the study to a single, large Pt particle raises the question of the statistical relevance of the observed twinning/detwinning. In situ TEM is also limited to a few clusters, but at least statistical averaging is typically done. 200 nm is also very large for a catalytic Pt particle. These limitations should be clearly discussed and data points for a few particles should be included if possible, maybe only at initial and final conditions.

Answer II: We agree that 200 nm is very large for a catalytic Pt particle. The studied particle can be considered as a model particle. Indeed, the catalytic activity of Pt particles is mainly associated with the defect sites, such as facet edges or the surface line defects associated with twin boundaries. The relative fraction of such sites is higher in smaller nanoparticles, yet the underlying physics is the same also in larger particles. It can be observed that even large particles (see the work of Abuin *et al.* [Abuin *et al.*, *ACS Appl. Nano Mater.* 2, 4818–4824 (2019)] or Kim *et al.* [Kim *et al.*, *Nat. Commun.* 9:3422 (2018)]) show strain or shape evolution during reaction. The observed twinning/detwinning phenomenon has been observed on one particle. During the synchrotron run, we did not have time to measure another particle with a twin defect (aligning and finding particles are time-consuming). However, we think that our results on a single particle showing twinning/detwinning are still relevant because of the observation of the chemical origin of the restacking forces.

Fig S3 suggest that detwinning is mostly a function of time, and not conditions. This seems logical considering the high energy of the twinning defect. Is CO adsorption driving twinning/detwinning as suggested by the DFT calculations? Or is the local heat of the CO oxidation reaction relevant? Is a control experiment possible?

Answer III: Figure S3 suggests that detwinning evolves linearly with time. From the DFT calculations, we expect CO adsorption to initiate the twinning process. We do not have evidence of local heat during the CO oxidation reaction. By measuring the 3D diffraction pattern of the particle, we have access to its average strain or average lattice d-spacing. No change of the average strain/d-spacing of the (111) planes (2.276 ± 0.001 Å) is observed, meaning that no heat can be evidenced by nano-focused 3D coherent diffraction imaging during the experiment.

Was activity measured during the experiment? It would be nice to complement the structural data.

Answer IV: The activity was measured during the experiment. It is now added in the new version of the manuscript (see Fig. 2).

The particle preparation procedure seems relevant – during preparation the particles were exposed to air and 800 C. How does this affect the resulting particles? How common are twinning defects in these particles?

Answer V: We agree that the particle preparation procedure (annealing at 800°C) is prone to the formation of twin and grain boundaries. The resulting particles are also less faceted than particles annealed at higher temperatures. Particles annealed at higher temperatures show less defects. In the Methods section, we added the sentence: “The low (800°C) annealing temperature is prone to the formation of twin and grain boundaries”.

The experiments show both strain and twinning/detwinning. The calculations only consider adsorption on the (511) and the (111) surface. Strain effects were not considered, though they are at least as important. Twinning is a bulk defect. How is this connected to surface calculations?

Answer VI: It will be very interesting to include strain effects. In the manuscript, they are not connected to the DFT (surface) calculations. Combining surface and 3D strain effects is an ambitious work, which goes far beyond this manuscript. As mentioned in the manuscript, we assume that strain has also an impact on the twinning/detwinning process: “In addition to the anisotropy of interfacial energy between free surfaces exposed to CO, the change of local strain/displacement at the surface of the particle during reaction may also contribute to the twinning process”. Moreover, the potential of strain energy alone to move the twin boundary was probably exhausted during annealing at the temperature of 450 °C for 2 h 30 min prior to BCDI measurements and CO exposure, as outlined in “Sample history” in “Methods” section.

The surface energy (gamma) is a free energy. The adsorption entropy has an important effect on the surface coverage and the stability and therefore needs to be included in the calculations.

Answer VII: We agree with the Reviewer that the gamma is a free energy and that the Gibbs free energy of adsorption should be included in the calculation of gamma. As reported in Refs. <https://pubs.acs.org/doi/10.1021/acscatal.9b01840> and <https://pubs.acs.org/doi/10.1021/acscatal.8b02371>, entropic effects are indeed important in describing the thermodynamics of CO adsorption onto Pt surfaces. However, it should be noted that differences in CO adsorption energies (i.e. binding energies) agree well with differences in Gibbs free energies of adsorption when different free surfaces of Pt are compared, as reported in Ref. <https://pubs.acs.org/doi/10.1021/acscatal.9b01840>. Thus, because differences in the interfacial energy between different surfaces as a function of the coverage is given by the differences in the clean surface and the differences in the Gibbs free energy of adsorption (or binding energy), we expect our approach to yield a good estimate of the relative difference in gamma. We have added a sentence in the method section to clarify this:

We note that the adsorption energy, rather than the Gibbs free energy of adsorption, is employed in equation (2). However, because differences in the Gibbs free energy of adsorption of CO computed for different free surfaces have been shown to be almost equal to the differences in the adsorption energies computed using equation (3) (Ref. <https://pubs.acs.org/doi/10.1021/acscatal.9b01840>), we are confident that our approach provides a good estimate of the difference between $\gamma_{111}^{CO}(n)$ and $\gamma_{511}^{CO}(n)$.

Finally, we note that (511) surface is more disordered than its (111) counterpart, and it is expected that it will also exhibit higher entropy. Thus we expect the difference in gammas at high CO coverages to be even higher than that shown in Fig. 5.

Why were only the (511) and (111) surface considered? The (100) surface is also highly stabilized by CO adsorption, as shown in earlier work.

Answer VIII: Only the (511) and (111) surfaces have been considered as no (100) surface was observed on the particle. They represent also the largest portion of the free surface of the particle.

REVIEWER COMMENTS

Reviewer #1 (Remarks to the Author):

Report on manuscript entitled „Twin boundary migration in an individual platinum nanocrystal during catalytic CO oxidation “ by Carnis et al.

The authors have addressed most of the points raised in my previous support successfully but they need to get the CO adsorption thermodynamics right before the paper can be published, because this has significant impact on the interpretation given in the manuscript.

This refers especially to answer 9. The conditions in the experiment correspond to 723 K and 2×10^{-2} bar CO (2% CO and reactor pressure approx. 1 bar). This corresponds to a max coverage of about 0.33 ML (see Fig. 5 in ACS Catal. 2018, 8, 10225–10233) and attached document.

The statement, that “However, because differences in the Gibbs free energy of adsorption of CO computed for different free surfaces have been shown to be almost equal to the differences in the adsorption energies computed using equation (3) (Ref.

<https://pubs.acs.org/doi/10.1021/acscatal.9b01840>), we are confident that our approach provides a good estimate of the difference between γ^{111} and γ^{511} .”

I disagree with this statement: exactly because the differences in adsorption energies for different surfaces are not the same as the differences in Gibbs free energy, the nanoparticle shape changes occur, as they are reported in the reference given.

To come to proper conclusions for their interpretation, the authors need to

- provide calculations using the same functional for adsorption structures on the 111 and the 511 surface
- draw the surface phase diagram for the Gibbs free energy as a function of the CO chemical potential for the calculated structures for both surface (this is straight forward when the adsorption energies have been calculated)
- read off the most stable surface structure and surface orientations as well as Gibbs free energy differences, which could act as driving force for the observed process
- adopt the manuscript conclusions accordingly
- discuss the result within the error bars of the calculations

Reviewer #2 (Remarks to the Author):

The authors have answered my questions in their resubmission.

Reviewer #3 (Remarks to the Author):

Despite the detailed response to the comments, some concerns remain:

1. the relevance for catalysis of observations for a single, very large Pt particle remains an issue. Are the conclusions general or are they only relevant for this specific very special particle?
2. since the rate of twin boundary migration does not depend on the reaction conditions (in Fig S3 the migration rate remains the same even when CO is removed from the gas phase), i am not convinced that the difference in CO adsorption energy is driving boundary migration.
3. by neglecting the adsorption entropy, CO adsorption is too strong and the equilibrium CO coverage is overestimated. As a result, the surface stabilization by CO adsorption is overestimated and the minimum in Figure 5 will shift to lower coverages where the difference between the 111 and the 511

surface is smaller. It is a small effort to include (experimental) gas phase entropies, and this should be done.

Title: Twin boundary migration in an individual platinum nanocrystal during catalytic CO oxidation

No. NCOMMS-21-02033-T

ANSWERS TO REFEREES

We submit a revised manuscript and a supporting document for Reviewers with all the corrections highlighted in red. All responses to Reviewers and corrections to the manuscript have been listed as follows one by one.

Reviewer #1:

The authors have addressed most of the points raised in my previous support successfully but they need to get the CO adsorption thermodynamics right before the paper can be published, because this has significant impact on the interpretation given in the manuscript. This refers especially to answer 9. The conditions in the experiment correspond to 723 K and 2×10^{-2} bar CO (2% CO and reactor pressure approx. 1 bar). This corresponds to a max coverage of about 0.33 ML (see Fig. 5 in ACS Catal. 2018, 8, 10225–10233) and attached document.

Answer: We thank the reviewer for this comment. In the new version of the manuscript, we have calculated the expected CO coverage. In order to establish the CO coverage expected for {511} and {111} at the experimental conditions of pressure and temperature, we have computed the Gibbs free energy of adsorption as a function of the CO chemical potential for the lowest-energy configurations, see Figs. 5a-b. We have added in the method section a description of how we have calculated the Gibbs free energy. At 700 K and at a pressure of 0.02 bar of CO, a coverage of 0.062 ML is predicted for the {111} facets, while 0.333 ML would adsorb onto the {511} facets, yielding a difference in γ_{hkl}^{CO} of 0.84 J/m². At 300 K and 0.02 bar, we predict a coverage of 0.56 ML for {111} (see Fig. S23), in good agreement with recent computational studies by Sautet *et al.* [Sumaria, V., Nguyen, L., Tao, F. F. & Sautet, P. Optimal Packing of CO at a High Coverage on Pt(100) and Pt(111) Surfaces. *ACS Catal.* 12 (2020)] and Gunasooriya *et al.* [Gunasooriya, G. T. K. K. & Saeys, M. CO Adsorption on Pt(111): From Isolated Molecules to Ordered High-Coverage Structures. *ACS Catal.* 8, 10225–10233 (2018)].

The statement, that “However, because differences in the Gibbs free energy of adsorption of CO computed for different free surfaces have been shown to be almost equal to the differences in the adsorption energies computed using equation (3) (Ref. <https://pubs.acs.org/doi/10.1021/acscatal.9b01840>), we are confident that our approach provides a good estimate of the difference between γ^{111} (n) and γ^{511} (n).” I disagree with this statement: exactly because the differences in adsorption energies for different surfaces are not the same as the differences in Gibbs free energy, the nanoparticle shape changes occur, as they are reported in the reference given.

Answer: We agree, but what we were saying was indeed the opposite *i.e.* that the difference in Gibbs free energies is the same as the difference in total energies, and we had reported the values taken from two publications. In any case, we have now computed the Gibbs free energy by including entropic effect of gas phase CO, which should be the most dominant contribution.

To come to proper conclusions for their interpretation, the authors need to
- provide calculations using the same functional for adsorption structures on the 111 and the 511 surface

Answer: We have used the same functional. All the results reported in the manuscript in Figure 5 are obtained using revPBE. We have further clarified this in the text. As explained in our previous response, we employed also rev-vdw-DF2 to show that the results do not depend on the choice of the functional used.

- draw the surface phase diagram for the Gibbs free energy as a function of the CO chemical potential for the calculated structures for both surface (this is straight forward when the adsorption energies have been calculated)

Answer: We thank the Reviewer, we have added in the new Figure 5 the plots of the Gibbs free energy of adsorption as a function of the chemical potential of CO to determine the coverage at the experimental conditions of pressure and temperature.

- read off the most stable surface structure and surface orientations as well as Gibbs free energy differences, which could act as driving force for the observed process
- adopt the manuscript conclusions accordingly

Answer: This has been added in Figure 5 and in the discussion.

- discuss the result within the error bars of the calculations

Answer: It is rather uncommon to report error bar from calculations possibly due to the arbitrariness in its determination. We stress however that the computed adsorption energy at low coverage are in good agreement with experimental values so we are confident in our choice of the functional. We have added a sentence in the manuscript in this respect.

Reviewer #3:

Despite the detailed response to the comments, some concerns remain:

1. the relevance for catalysis of observations for a single, very large Pt particle remains an issue. Are the conclusions general or are they only relevant for this specific very special particle?

Answer: The studied particle can be considered as a model particle. Indeed, the catalytic activity of Pt particles is mainly associated with the defect sites, such as facet edges or surface line defects associated with twin boundaries. The relative fraction of such sites is higher in smaller nanoparticles, yet the underlying physics is the same also in larger particles. It can be observed that even large particles (see the work of Abuin *et al.* [Abuin *et al.*, ACS Appl. Nano Mater. 2, 4818–4824 (2019)] or Kim *et al.* [Kim *et al.*, Nat. Commun. 9:3422 (2018)]) show strain or shape evolution during reaction.

We expect the conclusions to be general to particles with surface defects like twin boundaries. At 450°C, we stayed a very long time (2h30) under Ar without any change of the particle and when switching to CO, we directly observed the twin boundary migration. This implies that the twin boundary movement is initiated by chemical changes.

2. since the rate of twin boundary migration does not depend on the reaction conditions (in Fig S3 the migration rate remains the same even when CO is removed from the gas phase), i am not convinced that the difference in CO adsorption energy is driving boundary migration.

Answer: We agree that it is quite puzzling that the rate of twin boundary migration does not depend on

the reaction conditions but on time. Nevertheless, as shown experimentally, the twin boundary migration has been initiated under CO flow.

3. by neglecting the adsorption entropy, CO adsorption is too strong and the equilibrium CO coverage is overestimated. As a result, the surface stabilization by CO adsorption is overestimated and the minimum in Figure 5 will shift to lower coverages where the difference between the 111 and the 511 surface is smaller. It is a small effort to include (experimental) gas phase entropies, and this should be done.

Answer: We thank the Reviewer, we have indeed added the gas phase entropies to compute the Gibbs free energy and determined the coverage at 700 K and 0.02 bar. This result is reported in the new Figure 5 and the discussion of the gamma is now reported in the manuscript.